# Contribution of Infectious Agents to the Development of Celiac Disease

**DOI:** 10.3390/microorganisms9030547

**Published:** 2021-03-06

**Authors:** Daniel Sánchez, Iva Hoffmanová, Adéla Szczepanková, Věra Hábová, Helena Tlaskalová-Hogenová

**Affiliations:** 1Laboratory of Cellular and Molecular Immunology, Institute of Microbiology of the Czech Academy of Sciences v.v.i., Vídeňská 1083, 142 20 Prague, Czech Republic; szczepankova.adela@gmail.com (A.S.); habova@biomed.cas.cz (V.H.); tlaskalo@biomed.cas.cz (H.T.-H.); 2Department of Internal Medicine, University Hospital Královské Vinohrady and Third Faculty of Medicine, Charles University, Ruská 87, 100 00 Prague, Czech Republic; iva.hoffmanova@fnkv.cz; 3First Faculty of Medicine, Charles University, Kateřinská 1660/32, 121 08 Prague, Czech Republic

**Keywords:** celiac disease, infections, microbiota, parasites, gluten-free diet, immune response

## Abstract

The ingestion of wheat gliadin (alcohol-soluble proteins, an integral part of wheat gluten) and related proteins induce, in genetically predisposed individuals, celiac disease (CD), which is characterized by immune-mediated impairment of the small intestinal mucosa. The lifelong omission of gluten and related grain proteins, i.e., a gluten-free diet (GFD), is at present the only therapy for CD. Although a GFD usually reduces CD symptoms, it does not entirely restore the small intestinal mucosa to a fully healthy state. Recently, the participation of microbial components in pathogenetic mechanisms of celiac disease was suggested. The present review provides information on infectious diseases associated with CD and the putative role of infections in CD development. Moreover, the involvement of the microbiota as a factor contributing to pathological changes in the intestine is discussed. Attention is paid to the mechanisms by which microbes and their components affect mucosal immunity, including tolerance to food antigens. Modulation of microbiota composition and function and the potential beneficial effects of probiotics in celiac disease are discussed.

## 1. Celiac Disease—Introduction

The ingestion of wheat gliadin (the alcohol-soluble part of wheat grain storage proteins, i.e., gluten) and phylogenetically related cereal proteins (secalin from rye, hordein from barley, and avenin from oat) induces celiac disease (CD) in genetically susceptible individuals bearing the human leukocyte antigens HLA-DQ2 (alleles DQA1*0501 and DQB1*0201) and HLA-DQ8 (DQA1*0301 and DQB1*0302) haplotypes [1,2,3,4]. CD affects about 1:100–200 of those who consume wheat. Helper T-cells (Th1) are the dominant effector immunocytes driving the damage to the small gut mucosa, which is characterized by villous atrophy and crypt hyperplasia accompanied by malabsorption and gastrointestinal symptoms. Th1 cells activated by gliadin peptides, via antigen-presenting cells, produce high levels of pro-inflammatory cytokines (IL-2, IL-6, IFN-γ, and TNF-α) that promote increased cytotoxicity of intraepithelial lymphocytes and natural killer T cells, which in turn leads to damage of the small gut (villus) enterocytes (via abundant apoptosis) and simultaneously hyperplasia of crypts. Recruitment of Th2 cells during the immune response against gliadin leads to activation of B cells, their transformation into plasma cells (plasmacytes), and the production of antibodies against gliadin as well as autoantibodies, of which those against tissue transglutaminase and endomysium are considered to be serological hallmarks of a CD diagnosis (Figure 1). Life-long adherence to a gluten-free diet (GFD) is the sole rational therapy for CD, with the goal of healing pathological changes in the gut mucosa and suppressing the production of antibodies and autoantibodies [3,5,6,7,8,9]. CD is associated with several extra-intestinal phenomena, including dermatitis herpetiformis and ataxia [10,11,12]. Liver disorders, respiratory symptoms, and alveolitis have also been linked to CD [13,14,15]. This corresponds to the increased prevalence of repeated infections in CD [16,17]. Certain viral and bacterial infections occurring in early postnatal life have been significantly associated with the development of CD [18,19].

Although gluten is unquestionably the trigger for CD, indirect evidence suggests that microorganisms may also play an essential role in the pathogenesis of CD and can impact the onset, progression, and even clinical presentation of the disease. The evidence includes: (1) early, multiple gastrointestinal infections, the microbiome/dysbiome repertoire, early vaccinations, and consumption of antibiotics or proton pump inhibitors (which themselves can induce dysbiosis) can trigger CD [17,20,21,22]; (2) although a GFD reduces CD symptoms in most CD patients, it does not entirely restore the duodenal/jejunal mucosa to the level found in healthy individuals [23,24,25,26]; (3) onset of CD can occur in adults, i.e., years after the introduction of gluten into the diet [27]; (4) a more than 80% concordance among monozygotic, i.e., equally genetically predisposed twins in the development of CD, indicates that one’s genetic background, represented mainly by HLA-DQ2/DQ8 haplotype (and non-HLA genes), is necessary but insufficient for the development of CD [28,29,30]; and (5) intestinal dysbiosis probably exists in both patients with active CD and those with long-lasting adherence to a GFD [26,31,32,33]. There is an increase in the pathobionts *Clostridium* spp. and Enterobacteriaceae while potentially protective bacteria (*Bifidobacterium* spp. and the *Lactobacillus* group) are decreased in CD patients [32,34,35,36].

## 2. Bacterial Infections in Celiac Disease

Infectious diseases are associated with both increased morbidity and mortality in CD patients [37]. The increased susceptibility of CD patients to infections is probably explainable, even beyond the genetically determined aberration of immune system functions, by impaired nutritional conditions, malnutrition, deficiency of vitamin D, folic acid, B12, hyposplenism, and altered mucosal intestinal permeability [38,39,40,41,42,43].

It is assumed that recurrent infections increase the risk of CD development [44,45]. Children suffering from more than ten gastrointestinal (or respiratory) infections are at a higher risk of CD development compared to children with less than four infection events during the reference period [21,45]. Additionally, mucosal infections may contribute to the impairment of immune tolerance to gluten, leading to tissue damage in CD patients [46].

Infections, mainly those induced by *Clostridium difficile, Helicobacter pylori,* and *Streptococcus pneumoniae* (*Pneumococcus*), are frequently associated with CD. A significantly higher hazard ratio, linked to *Clostridium difficile,* was found in CD patients compared to controls. A higher incidence of *Clostridium difficile* infection in CD was found in a large-scale population-based cohort study involving more than 28 thousand CD patients and more than 141 thousand controls; the incidence of *Clostridium difficile* infection in the former group was estimated at 56/100,000 person-years in contrast to the incidence of 26/100,000 in controls (i.e., the general population and non-CD controls). Interestingly, the risk of *Clostridium difficile* infection was highest in the first 12 months after diagnosis (hazard ratio: 5.2, *p* < 0.0001) and remained high, for up to five years, compared to controls [47].

There is a risk of CD in children suffering from peptic ulcers: *Helicobacter pylori* infection was present in 63% of patients with CD and 44% of non-celiac peptic ulcers; however, the difference was not statistically significant. Of the entire ulcer group, 11% of CD patients with peptic ulcers were negative for *Helicobacter pylori* infection [48]. The *Helicobacter* and *Megasphaera* genera were highly abundant in duodenal biopsy samples from adult CD patients compared to first-degree relatives and controls [49]. However, the role of these bacteria in the pathogenesis of CD has not been completely elucidated. It is a matter of debate whether these bacteria elicit pathological changes in the small gut mucosa of the celiac patients or the presence of these bacteria, due to immune dysregulation or intrinsic conditions in celiac patients, favor colonization by these bacterial species [30].

Nonetheless, CD patients are at increased risk for the development of bacterial infections [50]. There are also rare reports on CD patients with respiratory diseases; a Swedish population-based 2006 study reported that the tuberculosis risk was 3–4 times higher in CD patients [51,52]. CD is associated with an increased risk of pneumococcal infection. Invasive *Streptococcus pneumoniae* (*Pneumococcus*) infection is a particularly dangerous co-morbidity in CD patients. *Streptococcus pneumoniae* is a causative agent of pneumonia, bacterial meningitis, and sepsis. One of the first comprehensive studies concerning pneumococcal infection was performed in England. The objective of the study was to determine the risk (rate ratio) of pneumococcal infection in patients with CD in a population in: (1) the Oxford region (1963–1999); and (2) the whole of England (1998–2003). The high rate of pneumococcal infections in CD patients persisted beyond the first year after the CD diagnosis. It should be noted that the pneumococcal vaccination was available at the time of the all-England study but not at the time of the Oxford study, which influenced study conditions [42]. A Swedish study by Röckert Tjernberg et al. [41] showed an increased risk (although statistically insignificant) of invasive pneumococcal disease in CD. The risk estimate was similar after considering comorbidities, socioeconomic status, and education level [41]. An increased risk of bacterial pneumonia, especially in children and young people with CD, was found by Simons et al. [40] and Canova et al. [53]. Interestingly, the risk of bacterial pneumonia was significantly increased before the CD diagnosis [40,53]. For this reason, pneumococcal vaccination in individuals at risk of developing CD is the most effective way to prevent streptococcal pneumonia. Preventive pneumococcal vaccination should be considered for CD patients between 15 and 64 years who have not received the pneumococcal vaccination series in childhood [40]. The beneficial effect of the vaccination in CD, however, depends on splenic function [54]. Studies have reported a high prevalence of respiratory tract infections in CD patients with hyposplenism. Hyposplenism, along with malnutrition and vitamin deficiency, is the leading cause of susceptibility to respiratory infections such as streptococcal pneumonia. Interestingly, hyposplenism in adult CD patients ranges 19–80% [40]. Hyposplenism is associated with a fourfold increase in the risk of fulminant and fatal septicemia from encapsulated organisms in patients with CD [55]. The British Society of Gastroenterology recommends routine pneumococcal vaccination (pneumococcal 13-valent conjugate vaccine, PCV13 followed, at least eight weeks later, by the 23-valent pneumococcal polysaccharide (PPSV23)) for all adult patients diagnosed with CD [50,52]. Defense against pneumococcal infection in CD patients is probably based on cellular mechanisms because complement changes after *Streptococcus pneumoniae* infection seem similar in children with and without CD. It is unlikely that complement contributes to increased sensitivity to invasive pneumococcal infection in these individuals [56].

Moreover, vulnerability to respiratory infections in CD patients may also be caused by impaired airway epithelium function, e.g., a defect in nasal mucociliary clearance, which has been found in CD patients. A defect in nasal mucociliary clearance increases the risk of lung infections. Nasal mucociliary clearance time was significantly prolonged in pediatric CD patients compared to healthy children. Interestingly, no relation was found between the age at diagnosis, histopathological stage, or compliance with a GFD relative to the nasal mucociliary clearance time in CD patients [57].

Adult CD patients were shown to be at significantly increased risk of sepsis (hazard ratio = 2.6), especially pneumococcal sepsis (3.9). A similar situation exists in CD children (1.8) [43]. Moreover, respiratory infections are closely related to the increased mortality associated with CD. The cause-specific mortality risks in the periods before and after introducing accurate and specific serological tests for the diagnosis of CD were studied by Grainge et al. [58]. CD mortality did not change significantly after introducing routine serological testing as part of the CD diagnosis [58]. The Swedish national mortality register, which includes approximately ten thousand CD patients, was analyzed to determine the cause of death in these patients. The cumulative mortality risk for cancer, digestive disease, and respiratory diseases (including pneumonia, allergic disorders, and asthma) was significantly elevated in CD patients [59].

## 3. Viral Infections Associated with Celiac Disease

Patients with CD were described to have elevated antibody levels to human adenovirus serotype 2, which indicates infection by this virus [60,61,62]. Kagnoff et al. suggested that molecular mimicry may play a pathogenic role, i.e., immunological cross-reactivity between antigenic epitopes of viral proteins and gliadins via shared antigenic determinants. These authors characterized a sequence homology between the E1b region of human adenovirus serotype 12, isolated from the human intestinal tract, and an A-gliadin (an alpha-gliadin component known to be harmful to the small intestine of CD patients) [60,61].

Generally, it was assumed that the Reoviridae family (mainly reoviruses and rotaviruses), adenoviruses, the respiratory syncytial virus, herpes simplex type 1, hepatitis C and B viruses, enteroviruses, the influenza virus, cytomegalovirus, and Epstein-Barr virus might play a role in CD development [19,63]. Reoviridae infections are common and usually nonpathogenic, although some viruses in this family, namely the rotaviruses, can cause severe diarrhea and abdominal discomfort in children [64]. High rates of rotavirus gastroenteritis in children with CD were found [16], and vaccination against rotavirus prevents the onset of CD [65].

Rotavirus infections have been proposed to trigger CD and type 1 diabetes mellitus in genetically susceptible children via molecular mimicry. When measuring the prevalence of diseases in a group of children vaccinated with RotaTeq (Kenilworth, NJ), a Finnish study showed that the prevalence of CD was significantly lower in vaccinated children than in those receiving a placebo [66]. However, the introduction of the rotavirus vaccination in Italy did not affect CD prevalence in Italian children [67]. Moreover, several viruses, namely rotaviruses and astroviruses, directly increase gut mucosal permeability [68]. Reoviruses promote enterocyte apoptosis and may trigger a pro-inflammatory response against ingested food antigens [69].

An essential role for reovirus infections in the pathogenetic mechanism of CD was recently suggested [69,70]. Using an animal model of human disease, reovirus was shown to disrupt intestinal immune homeostasis, promote immunopathology by suppressing peripheral regulatory T cells, and activate the pathogenic Th1 response, which led to the loss of immune tolerance to gliadin. The central role of IRF1 (interferon regulatory factor 1) in the reovirus-mediated Th1 response against dietary antigen was also demonstrated [69]. Anti-reovirus antibody titers were shown to be higher in a cohort of celiac patients consisting of patients with active CD and those on a GFD in contrast to control individuals. CD patients on a GFD with high anti-reovirus antibody titers possess significantly higher interferon regulatory factor 1 expression in the small intestinal mucosa compared to those patients with low-reovirus antibody titers. Nevertheless, there was no direct relationship between anti-reovirus antibody levels and the level of IRF1 expression. These results support the opinion that viruses may influence the transcriptional program of the host for long periods of time [70].

The association between CD and prior respiratory syncytial virus infection or viral bronchiolitis has been studied [39]. Approximately four thousand children suffering from CD (March III stage of villous atrophy) were statistically analyzed for respiratory syncytial virus infections or viral bronchiolitis before their CD diagnosis. Prior to the CD diagnosis, 0.9% of CD patients, in contrast to 0.6% of matched controls, were infected by the respiratory syncytial virus. The odds ratios were similar for girls and boys. Interestingly, the highest odds ratios were found in patients developing CD before one year of age. In this study, 3.4% of CD patients and 2% of matched controls had viral bronchiolitis, regardless of the viral agent [39].

Enteroviruses have been found in the small gut mucosa of CD patients [71]. Recently, early childhood exposures to enteroviruses, between the age of one and two years, was associated with an increased risk of CD [72].

The CD-associated DQA1*0501/DQB1*0201 haplotype causes susceptibility to herpes infection due to delayed maturation of the gastrointestinal immune system and mucosal overexpression of the epidermal growth factor receptor and IL-33 [73]. CD is an immune-mediated disease associated with a high rate of non-response to the viral hepatitis B vaccination. Proper CD treatment might lead to a positive response to the hepatitis B vaccination [74]. Interestingly, a significantly increased risk for the development of CD after vaccination with the quadrivalent human papillomavirus vaccine was found in a Danish and Swedish population study [75].

Influenza has also been linked to autoimmune conditions. Mårild et al. found that children with active CD were at increased risk for influenza (hazard ratio of 2.5) [38]. Analysis of the risk of CD after influenza in Norwegians revealed a significantly increased hazard ratio for CD after seasonal and pandemic influenza. The hazard ratio remained significantly increased one year after influenza, while the hazard ratio for influenza after CD diagnosis was not significant, although individuals with CD were at increased risk of hospital admission for influenza (2.1) [38,76].

A number of autoantibodies and antibodies against food antigens develop in patients with active CD in contrast to healthy individuals. Conversely, serum levels of IgG antibodies to cytomegalovirus and Epstein-Barr virus were lower in CD patients than healthy controls [19]. More recently, an inverse correlation was described among anti-cytomegalovirus, anti-Epstein-Barr virus, and anti-herpes simplex type 1 virus IgG antibody levels and levels of autoantibodies against tissue transglutaminase [77].

## 4. *Candida albicans* and Parasitic Protozoa in Celiac Disease

*Candida albicans* is a commensal of the human gastrointestinal tract that can convert into a pathogen. There are sequence similarities between *Candida albicans* hyphal-wall proteins (namely, hyphal wall protein 1, HWP1) and T-cell alpha-gliadin and gamma-gliadin epitopes in CD patients. Moreover, HWP1 is also a substrate for mammalian transglutaminases. Mammalian transglutaminases mediate adherence of *Candida albicans* to enterocytes via covalent bonding of its proteins to the host tissue [78]. Analysis of serum IgG antibodies against recombinant Hwp1 showed significantly elevated levels of these antibodies in patients with CD and patients with *Candida albicans* infections. There are no statistical differences in antibody levels in CD patients and patients with *Candida albicans* infections. Interestingly, elevated antibodies against Hwp1 paralleled elevated levels of IgA anti-gliadin antibodies in the course of infection with *Candida albicans*. These findings led the authors to hypothesize that *Candida albicans* infection may trigger CD onset in genetically susceptible individuals [79].

CD patients seem to be susceptible to invasion by the unicellular eukaryotic parasites (parasitic protozoa) *Toxoplasma gondii* and *Giardia lamblia*. An approximately four-fold higher risk for *Toxoplasma gondii* is seen in individuals positive for CD autoantibodies (autoantibodies against tissue transglutaminase), indicating faster development of *Toxoplasma gondii* oocyst in the gut of these patients [80]. Elevated autoantibodies against tissue transglutaminase and endomysium and pathological duodenal histology, during or shortly after a *Giardia* infection, have also been found [81].

## 5. Involvement of Microbiota in the Pathogenesis of Celiac Disease

### 5.1. Gut Microbiota in Celiac Patients

The maturation of the immune system’s innate and adaptive components provides a mechanism for distinguishing harmful from harmless substances on mucosal surfaces. A critical function of the mucosal immune system is connected with the establishment of mucosal (oral) tolerance to food and microbiota antigens. Microbiota seems to be a key player in the development of a functional immune system. On the other hand, microbiota components may contribute to the development of immunopathological events [82,83,84]. Changes in both the composition and characteristics of the gut microbiome may lead to chronic intestinal and extraintestinal diseases, including autoimmune CD. Dysbiosis probably accompanies the prodromal phase of CD as well as the disease course and manifestation. In CD, dysbiosis may impair mucosal components of immune cells, including epithelial cells, making them more vulnerable to infection and autoimmune responses. CD-associated dysbiosis is observable in feces specimens, duodenal biopsies, saliva, and oropharyngeal swabs [85,86,87,88,89,90,91]. *Neisseria flavescens*, *Proteobacteria*, *Staphylococcus epidermis, Staphylococcus haemolyticus*, *Staphylococcus aureus,* and *Escherichia coli* are abundant in the stools of CD patients [25,88,90,92,93,94]. *Proteobacteria* were more abundant in the duodenal biopsies of adult CD patients and those suffering from dermatitis herpetiformis (dermal manifestation of CD) than in healthy individuals. Intriguingly, the duodenum and oropharynx shared similar microbiome profiles in CD patients, characterized by an abundance of the *Proteobacteria* phylum and *Neisseria* species. Thus, there is probably a microbiome continuum in active CD. The increased numbers of *Actinobacteria* and the reduced presence of the *Bacteroidetes* and *Fusobacteria* phyla were also found in the oral microbiota of refractory CD patients [95].

The mucosa-associated microbiota in the proximal gut of children with CD is abundant in *Clostridium*, *Prevotella*, and *Actinomyces,* which contrasts with healthy individuals. Moreover, decreased numbers of *Lactobacillus* and *Bifidobacteria* (except *Bifidobacterium bifidum)* and increasing numbers of Gram-negative bacteria, mainly proteobacteria, were found in pediatric CD patients [96,97]. *Clostridium leptum*, *Bacteroides*, *Staphylococcus,* and *Escherichia coli* were significantly more abundant in both stool and duodenal biopsy samples of pediatric CD patients. In contrast, numbers of *Clostridium histolyticum*, *Clostridium lituseburense,* and *Faecalibacterium prausnitzii*, as well as *Bacteroides fragilis,* were higher in healthy individuals than in CD patients [87,88,90,93,95].

Based on in vitro experiments and the use of animal gnotobiotic models, it was shown that *Escherichia coli* could activate innate immune cells in the presence of gliadin, while *Bifidobacteria*, in contrast, possessed an inhibitory effect on these cells [87,88,93]. Beneficial commensal bacteria (such as *Bifidobacterium*) were shown to produce molecules preventing mucosal colonization by enterohemorrhagic *Escherichia coli* O157:H7 [98,99]. Beneficial gut microbiota, such as *Lactobacillus* sp. and *Akkermansia muciniphila,* contribute to the maintenance of the mucous layer (barrier) [100,101] and favor colonization resistance by competing for nutrients with pathogens, stimulating enterocytes to secrete antimicrobial molecules into the mucus, thereby supporting more effective protection against pathogens [102]. Intestinal commensal bacteria generally protect from infections caused by pathobionts via colonization resistance and interaction with the immune system. Colonization resistance is a complex set of events in which the indigenous microbiota (after microbial colonization of newborns) protects the host from invading pathogenic microbes with the potential to compromise the function of the mucosal barrier. Analyses of bacterial-derived small RNA could help elucidate the complicated relationship between microbiota and host relative to the pathogenesis of CD [103].

### 5.2. Metabolomics

Commensal bacteria harvest energy from food and produce metabolites of crucial importance to the host’s physiology, including development, polarization, and responsiveness of the immune system to invading microorganisms, as well as mediating protection against gastrointestinal infectious diseases. Furthermore, gut bacteria facilitate the digestion of insoluble fiber, produce vitamins such as vitamin K, and produce trophic and immunomodulatory compounds (e.g., short-chain fatty acids (SCFA)). There are several mechanisms by which primary or opportunistic pathogenic microbiota negatively affect the host immune system, leading to a breakdown of mucosal tolerance to gluten in individuals predisposed to CD. The declining numbers of *Lactobacillus* spp. are associated with decreased production of SCFA. SCFA possess a wide range of biological activities. They are necessary for correct maturation of the mucosal immune system [104]. SCFA can inhibit histone deacetylases, which prevents hyperacetylation of histones and can epigenetically modify the function of both innate and adaptive immune cells relative to the suppression of the inflammatory immune response [105,106]. SCFA, especially butyrate, promote the development, persistence, and activation of regulatory T-cells (T_reg_ cells) [107,108].

In terms of the metabolome analysis, only children who developed CD displayed elevated lactate production between 6–12 months of age, which correlated with high levels of *Lactobacillus* spp. in the gastrointestinal tract. Conversely, a decrease in *Lactobacillus* spp. was linked with a subsequent decrease in lactate production. Remarkably, decreasing lactate production occurred during the period crucial for the maturation of the mucosal immune system and establishment of immunological tolerance to food antigens (e.g., gliadin) and autoantigens. In infants genetically at risk for CD, early exposure (6–9 months) to gluten led to CD autoimmunity more frequently than those in which gluten exposure was delayed until 12 months of age [89,104]

### 5.3. Dysbiosis, Perturbation of Gut Mucosa Barrier, and Antigenic Load

Loss of integrity of the gut mucosa barrier can lead to an aberrant immune reaction against commensal microbiota, food antigens, and tissue antigens due to immune cross-reactivity, which may manifest as a failure of mucosal (oral) tolerance [109]. A disbalance in the microbiota can be a risk factor for CD since it negatively affects the integrity of the intestinal mucosa, which can lead to increased intestinal permeability, i.e., leaky gut [110,111]. Dysbiosis in CD is also associated with abnormal tight junctions and impaired barrier function, which leads to augmented translocation of gliadin peptides into the lamina propria and their exposure to the immune system. A “leaky gut” may assist with a harmful immune response to luminal (food) antigens. Indeed, broad immunological reactivity against food antigens and autoantigens is characteristics of active CD [112,113,114,115,116,117,118]. Certain bacteria can produce toxins that disrupt tight junction proteins and increase intestinal permeability, a condition linked to disruption of gluten tolerance. Increased intestinal permeability, both paracellular and transcellular, present in CD patients, contributes to the activation of adaptive immunity against gluten. Undeniably, *Bacteroides fragilis* expresses metalloproteases, which could participate in CD pathogenesis by this mechanism [119,120,121,122,123]. In general, the transit of food and microbiota antigens into the mucosa and submucosa can lead to hyperstimulation of the mucosal immune system, which may contribute to the impairment of gliadin tolerance. IgA antibodies, secreted by plasmacytes of gut-associated lymphoid tissue (GALT), represent an essential component of the gut mucosal barrier. They are part of the first line of defense against pathogens and participate in mucosal tolerance of harmless food and commensal microbiota components [82,124]. A reduction in the IgA-coated fecal bacteria in CD patients indicates the tendency to lose tolerance and decrease gut mucosa resistance [93]. Indeed, a reduction of IgA in the feces was shown to precede the onset of CD in infants [125].

Interestingly, the intestinal microbiota secrets transglutaminases, which catalyze the generation of neo-antigens/neo-epitopes on food, bacterial, and host proteins via their post-translational modifications, i.e., (de)amination, (de)amidation, and covalent cross-linking. Tissue transglutaminase (the principal autoantigen in CD) and its family members, the exogenous microbial transglutaminases, form multiple neo-epitopes on the transglutaminase-gliadin complex during their enzymatic activity (i.e., covalent cross-linking, deamidation, transamidation, GTP-binding/hydrolysis, and isopeptidase activities). The transglutaminases catalyze isopeptide bonds and cross-linking of amine groups containing the acyl acceptor lysine and acyl groups containing the acyl donor glutamine. The covalent cross-linking of gliadin and transglutaminase also seems to be a significant disruptor of intestinal permeability [126]. Moreover, the neo-epitopes formed in the transglutaminase-gliadin complex elicit both antibodies against this complex, by specific activation of B-cells, as well as increased numbers of harmful (autoreactive) T cells in CD patients [73,127,128].

The precise role of microbiota in the pathogenesis of CD is still unknown. It has been postulated (the “hygiene hypothesis”) that the increase in the incidence of CD could be linked to a decreasing number of infectious stimuli during early postnatal life [127,128,129]. There are several technical, methodical, conceptual, and interpretational problems and biases regarding microbiota analysis up to the present time. Moreover, a definition of a healthy microbiome is still missing [130]. Inconsistencies in the results of microbiota analyses in CD patients may be partly due to the heterogeneity of CD patients and controls used in studies and the techniques and specimens (duodenal or fecal samples) used in analyses [87,131]. Individuals employed as healthy controls were subjected to upper gastrointestinal biopsies to check for signs of gastrointestinal disease. An unusual or atypical microbiota may have colonized these individuals. On the other hand, it is a matter of contention if the microbes isolated from untreated CD patients induced the pathological mucosal changes seen in these patients or the inflammatory conditions specifically selected particular microbiota members for colonization [30]. Thus, crucial questions remain open, and a causal relationship between microorganisms and CD development needs to be demonstrated.

## 6. Influence of Genetic Background, Delivery Mode, and Newborn Feeding on the Microbiota of Celiac Patients

The dominance of the two major bacterial phyla, *Firmicutes* and *Bacteroides,* is established in childhood; in addition, *Actinobacteria*, *Proteobacteria*, *Fusobacteria,* and *Verrucomicrobia* are also frequently detected bacteria [132]. Generally, children with a genetic risk of CD showed a lack of bacteria from the phyla *Bacteroidetes* and *Actinobacteria,* alongside an abundance of *Firmicutes, Proteobacteria, Staphylococcus* spp. and *Bacteroides fragilis* [104]. The HLA genotype influences the early intestinal microbiota composition. An association has also been postulated between the HLA-DQ2/HLA-DQ8 genotype and the gut microbiota’s composition, suggesting that the microbiota can also be a predisposing factor for developing CD [133,134]. Prospective cohort studies on the microbiome in CD showed that the HLA genotype *per se* affects the gut microbiota of infants at genetic risk of CD. An abundance of *Bifidobacterium* spp. and *Bifidobacterium longum* were present in the gut microbiota of infants with the lowest HLA-DQ genetic risk of CD [89].

Individuals with a genetic predisposition for CD, i.e., those having HLA-DQ-2 or DQ-8, may be prone to intestinal dysbiosis. The feces of infants genetically predisposed to CD had increased numbers of *Firmicutes* (*Clostridium*) and *Proteobacteria* (*Escherichia* and *Shigella*) and decreased representation of *Actinobacteria* (*Bifidobacterium*) and *Bacteroidetes* compared to low-risk infants [104,134]. Stool samples from infants at genetic risk for CD at four and six months of age, who did not develop CD, showed increased bacterial diversity over time. An increasing abundance of *Bifidobacterium longum* was found in control children, while higher levels of *Bifidobacterium breve* and *Enterococcus* spp. were found in those children who developed CD [124].

The tendency to develop dysbiosis may start in early life and depends on the type of childbirth. Vaginal delivery is linked with colonization by *Lactobacilli, Prevotella, Bacteroides*, and *Bifidobacteria*. Cesarean section, which is considered a risk factor for developing CD in predisposed individuals, leads to colonization mainly by environmental and maternal skin bacteria and less by *Bacteroidetes* [135,136,137,138,139,140,141]. Interestingly, it seems that the specific microbiota that colonizes the placenta can occur in the meconium [142,143,144].

Breast-feeding exerts a protective effect against infections and reduces the risk of CD development [145]. Breast vs. formula feeding also seems to be an essential factor in determining the possible development of dysbiosis in children, especially in children at risk for CD. Human milk oligosaccharides (HMOs) promote the growth of beneficial commensals (e.g., *Bifidobacteria*), prevent the growth of pathobionts (*Clostridium difficile*), and protect cell components of the mucosal barrier against harmful reactivity [146,147]. The pathogenic bacteria *Clostridium perfringens* and *Clostridium difficile* were found in the feces of children on formula feeding in contrast to breastfed infants [89,125]. *Clostridium perfringens* and *Clostridium difficile* (and *Clostridium* spp. generally) have been associated with formula feeding, while the presence of enterotoxic *Escherichia coli* has been associated with the HLA-DQ2 genotype [133].

Nevertheless, a higher prevalence of enterotoxigenic *Escherichia coli* was seen in children at high genetic risk for CD compared to infants with an intermediate genetic risk irrespective of whether they were breast-fed or formula-fed. The PROFICEL project showed that breast-feeding promotes colonization with the *Clostridium leptum* group and *Bifidobacterium* species (mainly *Bifidobacterium longum* and *Bifidobacterium breve*), whereas formula-feeding promoted colonization with the *Clostridium coccoides–Eubacterium rectale* group [133]. Enterotoxigenic *Escherichia coli* were found more often in the stool of breast-fed infants at genetic risk for CD within the first week of life and four and six months of age in contrast to breast-fed infants with a low or intermediate risk [89,125]. Breast-fed infants at high genetic risk for the development of CD showed a reduction in *Bifidobacterium* spp. and had a higher relative abundance of Proteobacteria and unclassified *Enterobacteriaceae* [133,134]. Interestingly, at four months of age, infants with the highest risk of CD development showed a higher prevalence of enterotoxigenic *Escherichia coli*, irrespective of milk feeding practices [89,125]. The duration of breastfeeding correlated with the occurrence of several bacterial genera in the gastrointestinal tract, of which especially *Lactobacillus* and *Bifidobacterium* were previously linked with positive health outcomes [148].

## 7. Microbiota and Gluten-Free Diet in Celiac Patients

Compliance with a GFD only leads to a partial restoration of the gut microbiota. Though recolonization by *Enterobacteria* or *Staphylococci* has been observed, fewer *Bifidobacteria* and *Lactobacilli* and increased numbers of *Bacteroides*, *Enterobacteriaceae,* and virulent *Escherichia coli* persisted in CD patients on a GFD [149]. *Escherichia coli* clones isolated from active and non-active CD patients carry more virulent genes than the species isolated from healthy controls [90,94]. Indeed, full microbiota restoration is not reached in CD patients on a GFD due to reduction in prebiotics [150] and intestinal dysbiosis exists both in patients with active CD and those with long-lasting adherence to a GFD; a GFD leads to a reduction in both potentially pathogenic microorganisms and beneficial *Bifidobacteria.* Nevertheless, besides a genetic background (i.e., mainly HLA-haplotypes), a GFD may contribute to dysbiosis in CD patients [151,152]. No significant differences in nutritional intake, either before or after the introduction of a GFD, were observed in CD patients; there was a reduction in saccharide (e.g., fructans) intake in CD patients on a GFD. Fructans possess prebiotic properties and are an energy source for gut commensal microbiota [153]. Generally, a reduced intake of polysaccharides and oligosaccharides with prebiotic properties, which act as an energy source for commensals in the gut microbiota, may aggravate gut dysbiosis in CD patients on a GFD. Reduced fructans intake could explain the decrease in beneficial gut bacteria seen in these patients [154]. Interestingly, patients who underwent GFD complaining persistent CD symptoms have a microbiota which resembles that of patients with irritable bowel syndrome [150,155].

## 8. Anti-Infectious Responses in Celiac Patients

Autoimmune and immunologically-mediated diseases are characterized by chronic tissue damage and functional impairment, which is caused by the reactivity of the host immune system. Loss of self-tolerance (tolerance to autologous antigens) is a necessary condition for developing such diseases. It has been described that pathogenic autoimmune reactions might be provoked by any number of microorganisms and their components [83,156,157,158,159,160,161]. Infectious stimuli, including viral and bacterial superantigens, stress proteins, and chaperones, may evoke an abnormal antigen presentation, activation of immune cells, overexpression of genes in MHC locus, i.e., classical and non-classical human leukocyte antigens (HLA-A-C, HLA-E, CD1b, CD1d, and MIC-A/MIC-B), and cross-presentation of antigens on antigen-presenting cells or ectopic presentation of tissue antigens [83,162,163,164,165,166,167,168]. The abnormal presentation of antigens (including autoantigens) and cell hyperstimulation may lead to inadequate as well as adverse reactions of the immune system. A vigorous immune response against microbial antigens boosted by their adjuvant effect may elicit activation of potentially autoreactive or cross-reactive juxtaposed immune cells (i.e., “bystander activation”) and recognizing autoantigens via “molecular mimicry” [169,170,171,172,173,174,175]. An aberrant intestinal immune response against pathogenic and commensal microbiota may trigger the development of immunologically-mediated diseases via this mechanism. The crucial role of the microbiota in the induction of immunologically-mediated diseases was confirmed by findings in experimental animal models of human diseases in our laboratory (inflammatory bowel disease, rheumatoid disease, psoriasis, uveoretinitis, inflammatory bowel disease, allergies), i.e., these diseases do not develop in germ-free conditions [84,176,177,178,179,180,181,182].

Immunocytes recognize the common structure of microbes (microbe-associated molecular patterns, MAMPs) via germline-encoded, phylogenetically old, pattern recognition receptors (PRRs). Toll-like receptors (TLRs), retinoic acid-inducible gene-I-like (RIG-I) receptors (RLRs), and nucleotide-binding oligomerization domain-like (NOD) receptors (NLRs) are among the most important members of the PRR family. Increased sensitivity to MAMPs suggests increased expression of TLR2, TLR4, and TLR9 in untreated and treated CD patients in contrast to controls [183]. Independently of HLA-DQ2 and DQ8, genes associated with protection against infection, such as *SH2B3*, have also been linked to CD. The *SH2B3* genotype-determined response to lipopolysaccharide and muramyl dipeptide reveals components of bacterial cells. The *SH2B3* rs3184504*A allele determines a stronger activation of NOD2 (Nucleotide-binding oligomerization domain-containing protein 2, an intracellular pattern recognition receptor, which upon activation trigger innate inflammatory responses) during defense against pathogenic bacteria [184]. Moreover, increased expression of TLR-9 mRNA and IL-8 mRNA (a pro-inflammatory cytokine) was observed in untreated CD patients compared to controls. TLR-9 is in the endoplasmic reticulum of plasmacytoid dendritic cells and early endosomes or late endosomes/lysosomes of the majority of innate immunity cells as well as T and B lymphocytes. TLR-9 recognizes non-methylated CG motifs (CpG oligodeoxynucleotides). Non-methylated CG motifs are common in bacterial and viral genomes but infrequently occur in mammalian DNA, except for mitochondrial DNA. Nevertheless, TLR-9 also recognizes Danger-associated molecular patterns (DAMPs). Thus, activation of the innate immune response, via TLR-9, might participate in the defense against infection, but also in the autoreactive response [185].

On the other hand, the expression of TLR-2 mRNA was significantly decreased in both untreated CD and CD-GFD patients compared to controls. Probably, the compensatory expression of inhibitory TOLLIP (Toll interacting protein; a negative regulator of TLRs) mRNA was also decreased in untreated CD patients compared to controls [186,187]. TLR-2 is localized predominantly on monocytes/macrophages, polymorphonuclear cells, dendritic cells, endothelial cells, hepatocytes, and activated B cells in the germinal centers of the tonsils, lymph nodes, and appendix. TLR-2 recognizes various bacterial, fungal, viral, and parasitological PAMPs (such as bacterial peptidoglycan, lipoproteins, lipoteichoic acids, lipoarabinomannan, porins, viral hemagglutinin, and glycoproteins) as well as several DAMPs [188,189,190]. Nevertheless, a reduction in TOLLIP expression may disrupt “lipopolysaccharide tolerance” in CD patients (along with an increased expression of TLR4), which prevents superfluous activation of immunocytes by microbial products, including lipopolysaccharide or lipoteichoic acid [187,191].

Cytokine IL-17A is produced by both CD8+T cells (Tc17 cells) and Th cells during inflammation and in anti-bacterial immune responses, as well as in active CD. *Prevotella* species, *Lachnoanaerobaculum umeaense,* and *Actinomyces graevenitzii,* have been isolated from CD jejunal biopsies. These species can trigger an IL-17A mediated immune response to gliadin peptides in some patients with active CD, as seen in experiments with intestinal biopsies from these patients [192,193].

The proinflammatory status of the intestinal mucosa of patients with active CD may be promoted by the presence of virulence genes encoding proteases in *Neisseria flavescens* and *Staphylococcus epidermis*. The expression of these genes may contribute to increasing the cleavage of gluten and, thus, the production of its immunogenic peptides in the duodenal mucosa of CD patients [24,25,88,90,92,93,94]. An abundance of virulence genes was also found in *Staphylococcus* spp., *Escherichia coli*, *Bacteroides fragilis* [90,94], and *Neisseria flavescent* [92] isolated from CD patients compared to controls. Moreover, the gene system for iron acquisition (genes for hemoglobin receptor, haptoglobin-hemoglobin A/B and transferrin A/B binding protein) differed between CD and control-associated *Neisseria flavescent*. *Neisseria flavescent* isolated from active CD patients possesses the gene for a hemoglobin receptor, usually present in pathogenic *Neisseria meningitidis*. Interestingly, *Neisseria* species isolated from active CD patients lacked the genes for haptoglobin-hemoglobin A/B and transferrin A/B binding protein, which are usually present in *Neisseria flavescent* strains [92].

A pivotal event in developing an immune response, both pro-inflammatory and tolerogenic, involves antigen-presenting cells, mainly dendritic cells. Five different strains of *Neisseria flavescens*, which were isolated from adult patients with untreated CD, were able to activate human dendritic cells and create an inflammatory phenotype [92]. The capacity of *Neisseria flavescens* to stimulate the host immune system could be related to the presence of virulence genes (e.g., bifunctional autolysin (*atl*E) and the methicillin-resistant gene (*mec*A)). These virulence genes are also found in *Proteobacteria* such as *Pseudomonas* and opportunistic pathogens such as *Staphylococcus epidermis,* which has been found in stools of CD patients [24,25,88,90,92,93,94].

Involvement of infections and dysbiosis in pathogenesis of celiac disease is schematically depicted in Figure 2.

## 9. Microbiota Modulation in Celiac Disease

Genetic background, type of delivery, maternal and infant nourishment, and antibiotic exposure can influence microbiota composition [136,194]. The symbiotic relationship between the host and colonizing microbiota establishes a selective immune tolerance to beneficial microbiota [195]. Dietary changes, antibiotics, and infections can disrupt intestinal microbial homeostasis and lead to increased intestinal permeability, i.e., leaky gut and gut dysbiosis [196]. Gut dysbiosis may contribute to the pathogenesis of CD and the persistence of CD symptoms in some CD patients on a GFD. These facts indicate the need to use probiotics as part of the treatment of CD in conjunction with a GFD [197]. The protective effects on the gut barrier function of probiotic strains such as *Lactobacillus casei* DN-114001 and *Escherichia coli* Nissle 1917 have been described [198]. In vitro experiments with Caco-2 cells found that the probiotic bacteria *Bifidobacterium lactis* protected epithelial integrity [199]. As already mentioned, several viruses, namely astroviruses and rotaviruses, increase gut mucosal permeability [68]. Despite the imperfect understanding of the precise mechanism regarding how the gut microbiota suppresses viral infections, oral administration of *Lactobacilli* spp. (*reuteri*, *acidophilus*, *delbrueckii*, *bulgaricus*, *casei*) and *Saccharomyces boulardii* have been shown to reduce the duration of rotavirus-associated diarrhea and substantially reduced viral shedding [200,201].

Thus, the utilization of probiotics could benefit CD patients. Generally, probiotics exert positive effects through prophylaxis against dysbiosis, e.g., caused by antibiotics. A study on Danish and Norwegian children showed that exposure to systemic antibiotics in the first year of life was associated with a later diagnosis of CD, which means antibiotics could be a risk factor for CD. The risk of antibiotic consumption relative to the development of CD was shown to be dose-dependent, i.e., the risk increases with the quantity of dispensed antibiotics. However, no specific type of antibiotics or age period (within the first year) was found to be crucial relative to the risk of developing CD [17,20,22,202,203].

Several studies analyzed the possible influence of various probiotics in CD patients. *Bifidobacterium infantis* in the form of Natren Life Start (NLS-SS) was administered to active CD patients. Administration of probiotics was found to lead to a significant improvement in gastrointestinal symptoms [204]. In contrast to the positive effects of probiotics mentioned, *Bifidobacterium infantis,* when administered to active CD patients, decreased the number of Paneth cells and the expression of alpha-defensin-5, as seen in duodenal biopsies [204,205]. Paneth cells are crucial in gut homeostasis and the regulation of innate immunity. These cells play an essential role in defense against pathogens by secreting defensins, lysozyme, and phospholipases [206].

Finally, drugs derived from microbial enzymes capable of degrading toxic gliadin peptides or probiotics bearing these enzymes are needed for complementary therapy of CD because absolute avoidance of gluten within a GFD is challenging to achieve [207,208]. In vitro (e.g., during gluten-free food preparation), degradation of gliadin pathogenic (proinflammatory/toxic) peptides is possible using proteases produced by *Bifidobacteria*, *Rothia* spp., *Actinomyces odontolyticus*, *Neisseria mucosa,* and *Capnocytophaga sputigena* [209,210]. Gluten-degrading microorganisms naturally occurred in the oral cavity, e.g., *Rothia mucilaginosa* and *Rothia aeria* [210,211], and *Lactobacilli* spp. [86].

In conclusion, many studies examining the role of microbiota in CD pathogenesis have been published. Studies suggesting the involvement of infections and microbiota in pathogenesis of celiac disease were focused on: (1) susceptibility of CD patients to infections [16,38,47,48,49,53,58,59,76]; (2) immunomodulatory effects of infections in CD patients and CD-susceptible individuals [37,44,45,60,61,62,68,69,70,72,76,80,84,120,158,160,161,168,171,172,173,174,175,191]; (3) dysbiosis in pathogenesis of CD [35,36,89,133,134,150,155]; (4) GFD and its influence on microbiota composition [26,149,151,152,153]; and (5) possible microbiota modulation in CD patients [98,100,137,148,154,197,198,199,200,204,205,209,210]. Nevertheless, further research is still needed. It is necessary to note that the exact role of the microbiota in the pathogenesis of CD is still not entirely unclear. Moreover, technical, methodical, conceptual, and interpretational problems and prejudices continue to influence microbiota research related to CD. At the same time, the definition of a healthy microbiome is still absent. Lastly, there is still an urgent need to identify new probiotic strains and bacterial proteases, which can cleave gliadin, that can be used in clinical praxis to improve the treatment of CD.

## Figures and Tables

**Figure 1 microorganisms-09-00547-f001:**
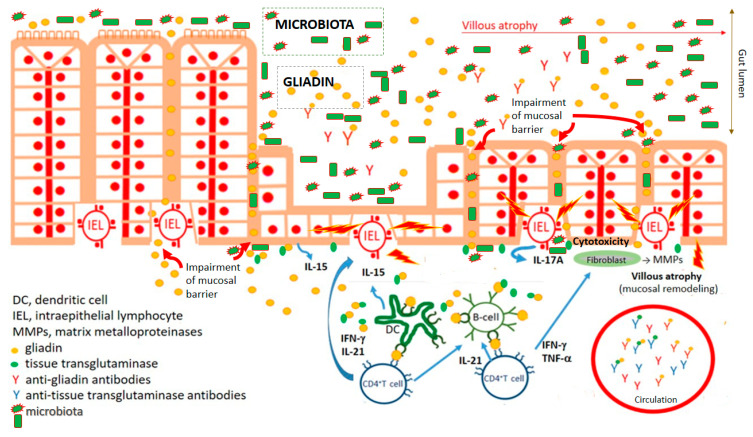
Key characteristics of celiac disease pathogenesis. Impairment of mucosal barrier of small intestine and penetration of food antigens, including wheat gliadin. Gliadin fragments are deamidated by tissue transglutaminase. Deamidated gliadin fragments stimulate (boosted by adjuvant properties of microbiota) innate immune cells, counting professional antigen presenting cells (dendritic cells), which present deamidated gliadin peptides by CD4^+^T cells. Polarization of CD4^+^T lymphocyte development to Th1 cytokine profile leads to activation (intraepithelial) lymphocytes and damaging of enterocytes. Activation of fibrocytes by Th1 cytokines trigger releasing of matrix metalloproteinases mediating pathological remodeling of small gut mucosa of celiac patients. Simultaneously, antibodies against gliadin and autoantibodies against tissue transglutaminase are developed.

**Figure 2 microorganisms-09-00547-f002:**
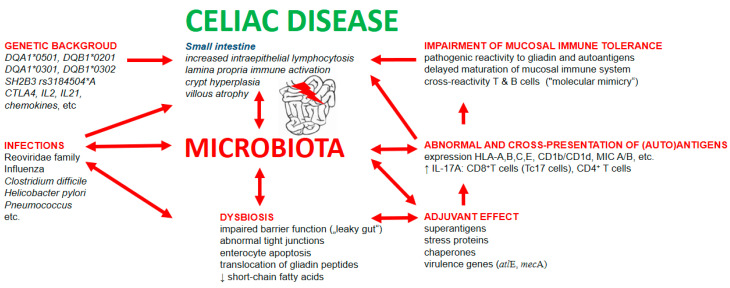
Involvement of infections and dysbiosis in pathogenesis of celiac disease. Infectious components possess the capability to non-specifically stimulate the immune system via: (1) adjuvant properties of bacterial and viral molecules; (2) stimulation of expression of classical and non-classical MHC molecules; (3) stimulation of antigen presentation, including cross-presentation of gliadin and autoantigens; (4) capacity of cross-reactivity of adaptive immune cells induced by mechanism of “molecular mimicry”; and (5) disruption of gut mucosal barrier leading to increased intestinal permeability for luminal food and bacterial antigens and thus elevated antigenic load in mucosal layer of small intestine.

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
