# Peer review of "Contribution of Infectious Agents to the Development of Celiac Disease"

_microorganisms, 2021, doi:10.3390/microorganisms9030547_

Round 1
Reviewer 1 Report
Sánchez et al. reviewed contibution of infectiou agents to development of celiac disease. Overall, this manuscript is well written and provide with a logical flow of mechanisms that in involved in disease progression. I have some minor comments.
- Some references are missing end of the sentences. It is not quite clear whether there is only one reference that refers to all paragraph, or it is just has been forgotten e.t.c. page 5; lines 178-179-181-189. It would be good to check all manuscript to avoid the confusion.
Author Response
Dear reviewer,
Thank you very much for your comments and requirements, which will ameliorate the manuscript quality. We have tried to meet of your requirements, i.e. we added the missing references and eliminated errors. All changes in text are indicated by underlining.
We hope that the revised manuscript will be now acceptable for publication in Microorganisms.
Thank you.
On behalf of all authors
Daniel Sánchez

Reviewer 2 Report
I've read with attention the paper of Daniel Sánchez et al. In this review, the authors widely discussed the relationship between bacterial, viral infections and gut microbiota dysbiosis and CLD development, as well as the mechanisms linking microbes and their components with mucosal immunity, including tolerance to food antigens.
The manuscript is logically organized and well written in English. The subject of the paper is sufficiently described and the authors referring to several research studies.
I have only a small suggestion: if possible, I would suggest authors add a Figure embracing the main findings in the context of CLD and bacterial/viral infections/gut microbiota state (and factors involving in gut microbiota modulation in CLD). This Figure will help readers to have a visual view of what is known so far in this field (like a graphical abstract).
Author Response
Dear reviewer,
Thank you very much for your comments and suggestion, which will ameliorate the manuscript quality. We have tried to meet of your requirement, i.e. we prepared two Figures depicting possible contribution of infectious agents in development of celiac disease. All changes in text are indicated by underlining.
We hope that the revised manuscript will be now acceptable for publication in Microorganisms.
Thank you
On behalf of all authors
Daniel Sánchez

Reviewer 3 Report
I reviewed with interest the present manuscript. Before a possible acceptance, the following point s should be addressed.
- I suggest abbreviating celiac disease with (CD)
- Recent evidence strengthen the concept of dysbiosis in CD (please see and amend 10.1111/jgh.15183 and 10.1053/j.gastro.2020.08.007
- Within 'Involvement of microbiota in the pathogenesis of celiac disease' section you should divide this part according to the following sub-section to improve clarity: 1. gut microbiota in CD patients 2. metabolomics in CD patients.
- Section: Gut microbiota in CD treated with GFD: the authors should better underline the microbial shift happening after GFD, highlighting that being this diet reduced in prebiotics, a full microbial restoration is not reached; moreover, patients who underwent GFD complaining persistent symptoms has a microbiota which resembles that of patients with IBS (Wacklin AJG 2015, Marasco AJG 2015).
- Microbiota modulation in CD: a recent comprehensive review was focused on this topic, which need to be mentioned ( doi: 10.3390/nu12092674).
- Minor English spelling errors
- The review should be enriched with figures showing the putative mechanisms of microbiota interaction with CD pathogenetic specific pathways.
- Table summarizing studies within microbiota and CD are needed
Author Response
Dear reviewer,
Thank you very much for your comments, suggestions and requirements, which will ameliorate the manuscript quality. We have tried to meet all of your requirements point-by-point. For this reason, we have: 1) added the new references, 2) prepared two Figures depicting possible contribution of infectious agents in development of celiac disease, 3) divided the section 5 (Involvement of microbiota in the pathogenesis of celiac disease) into sub-sections: 5.1. Gut microbiota in celiac patients, 5.2. Metabolomics and 5.3. Dysbiosis, perturbation of gut mucosa barrier and antigenic load. At your requirements, we replaced the abbreviation CLD with CD in the text of manuscript. The summarization of studies related to possible contribution of microbiota and infectious components in pathogenesis of celiac disease has been added in the last section of manuscript. All changes in text are indicated by underlining.
We hope that the revised manuscript will be now acceptable for publication in Microorganisms.
Thank you very much.
On behalf of all authors
Daniel Sánchez

Round 2
Reviewer 3 Report
The authors adequately responded to questions and modified the manuscript improving its quality.